# Biological Control of Tephritid Fruit Flies in the Americas and Hawaii: A Review of the Use of Parasitoids and Predators

**DOI:** 10.3390/insects11100662

**Published:** 2020-09-25

**Authors:** Flávio R. M. Garcia, Sérgio M. Ovruski, Lorena Suárez, Jorge Cancino, Oscar E. Liburd

**Affiliations:** 1Departamento de Ecologia, Instituto de Biologia, Zoologia e Genética, Universidade Federal de Pelotas, Pelotas 96010900, RS, Brazil; 2LIEMEN, División Control Biológico de Plagas, PROIMI Biotecnología, CCT NOA Sur-CONICET, Avda, Belgrano y Pje, Caseros, San Miguel de Tucumán T4001MVB, Tucumán, Argentina; sovruski@conicet.gov.ar; 3Dirección de Sanidad Vegetal, Animal y Alimentos de San Juan, Av. Nazario Benavides 8000 Oeste, Rivadavia CP 5400, San Juan, Argentina; ldcsuarez@sanjuan.gov.ar; 4Programa Moscafrut SAGARPA-IICA, Camino a los Cacahoatales s/n, Metapa de Dominguez 30860, Chiapas, Mexico; jorge.cancino.i@senasica.gob.mx; 5Entomology and Nematology Department, University of Florida, 1881 Natural Area Dr., Gainesville, FL 32611-0620, USA; oeliburd@ufl.edu

**Keywords:** classic biological control, conservation biological control, augmentative biological control, biological control programs

## Abstract

**Simple Summary:**

Biological control has been the most commonly researched control tactic within fruit fly management programs, and parasitoids have been the main natural enemies used against pestiferous fruit fly species. In view of this fact, it is important to highlight and compile the data on parasitoids with a certain frequency, aiming to facilitate the knowledge of all the researchers. Information regarding the activities of parasitoids and predators on pestiferous fruit flies in the Americas is limited; therefore, this study aimed to compile the diversity of parasitoids and predators associated with tephritid fruit flies, as well as providing the scientific evidence about the use of parasitoids and predators as biological control agents for fruit flies im the Americas and Hawaii.

**Abstract:**

Biological control has been the most commonly researched control tactic within fruit fly management programs. For the first time, a review is carried out covering parasitoids and predators of fruit flies (Tephritidae) from the Americas and Hawaii, presenting the main biological control programs in this region. In this work, 31 species of fruit flies of economic importance are considered in the genera *Anastrepha* (11), *Rhagoletis* (14), *Bactrocera* (4), *Ceratitis* (1), and *Zeugodacus* (1). In this study, a total of 79 parasitoid species of fruit flies of economic importance are listed and, from these, 50 are native and 29 are introduced. A total of 56 species of fruit fly predators occur in the Americas and Hawaii.

## 1. Introduction

In the Americas, there are four genera of tephritid fruit flies, which include species of economic and quarantine importance. *Anastrepha* Schiner, 1868 is widely distributed in the neotropical region [1], and seven species are economically important in the tropics and subtropics due to their wide range of commercial host plants and distribution. The species include *Anastrepha ludens* (Loew) (Mexican fruit fly), *A. obliqua* (Macquart) (the West Indian fruit fly), *A. fraterculus* (Wiedemann) (South American fruit fly), *A. suspensa* (Loew) (Caribbean fruit fly), *A. serpentina* (Wiedemann) (Sapotaceas fly), *A. striata* Schiner (guava fly), and *A. grandis* (Macquart) (melon fly). *Rhagoletis* Loew is mainly distributed in the neartic region and involves several pestiferous species such as *Rhagoletis pomonella* Walsh, one of the main pests of hawthorn in Central Mexico and apples in the Eastern United States, and *R. mendax* Curran, one of the most important blueberries pests in the Eastern United States [2]. *R. cingulata* (Loew) and its related *R. fausta* Osten Sacken are key pests of cherries, *Prunus* spp., in Eastern and Midwestern United States [3]. The native host of *R. cingulata* is the wild black cherry, *P. serotina* Ehrh, whereas *R. fausta* infests pin cherry, *P. pennsylvanica* L. [4]. A distant relative, *R. completa* (Cresson), infests walnuts and cause economic damage in the Western United States. There are also several neotropical *Rhagoletis* species associated with the Solanaceae, among which are two pestiferous species, *R. tomatis* Foote and *R. nova* (Schiner), which are associated with the tomato and the sweet cucumber, respectively [2].

*Bactrocera* Macquart and *Ceratitis* Macleay are two tephritid genera that were introduced into the continent. Two *Bactrocera* species are found in the Americas: *B. oleae* (Rossi) is present in the United States in the state of California [5], while *B. carambolae* Drew and Hancock is located in South America, distributed in Surinam, French Guiana, and Northern Brazil [6]. *Ceratitis capitata* (Wiedemann) is the only *Ceratitis* species in America, and it is widely distributed in Central and South America [7]. 

In Hawaii, *C. capitata* and three invasive *Bactrocera* species, i.e., *Zeugodacus cucurbitae* (Coquillett) (melon fly), *Bactocera dorsalis* (Hendel) (oriental fruit fly), and *Bactrocera latifrons* (Hendel) (Solonaceous fruit fly), are economically important and they rank high on quarantine lists worldwide [8].

Several natural enemies have been associated with tephritid fruit flies in the Americas and Hawaii either naturally occurring or through laboratory, field cages, and open-field testing with introduced or native species [9,10,11,12,13,14]. Biological control has been the most commonly researched control tactic within fruit fly management programs, and parasitoids have been the main natural enemies used against pestiferous fruit fly species [8,13]. In view of this fact, it is important to highlight and compile the data on parasitoids with a certain frequency, aiming to facilitate the knowledge to facilitate research in this field. The most recent and comprehensive review on the biological control of fruit flies using parasitoids was conducted 20 years ago covering Latin America and southern regions of the United States [11]. Since then, the main articles that reviewed augmentative biological control using parasitoids were from Mexico [15], Argentina [16], and Brazil [17]. Both American-native and exotic parasitoids are currently considered as candidates for biological control programs in the Americas [13,18,19,20]. The most recent reviews of parasitoids used in biological control programs in Hawaii were published by Bokonon–Ganta et al. [21] and Vargas et al. [8].

The use of entomopathogenic fungi and nematodes against fruit flies has shown promising results, which were recently reviewed by Dias et al. [13]. However, predators have been underexplored as control agents of fruit flies [11,22,23], although they are potentially important in conservation biological control [24,25]. Fruit fly immature stages are naturally exposed to a variety of predators, among which the most important are ants (Formicidae) and rove beetles (Staphylinidae), which have been detected to cause significant mortality [22,23,25,26]. Therefore, information on predator interactions with other biological control agents is the basis of studies on intraguild predation in augmentative biological control [27] and the compatibility between augmentative and conservation biological control in an approach for integrated pest management.

In this work, 29 species of fruit flies of economic importance are considered. They belong to the genera *Anastrepha* (12), *Rhagoletis* (10), *Bactrocera* (5), *Ceratitis* (1), and *Zeugodacus* (1), and are the following: *A. curvicauda* (Gerstaecker), *A. distincta* Greene, *A. fraterculus* (Wied.), *A. grandis* (Macquart), *A. ludens* Loew, *A. obliqua* (Macquart), *A. pseudoparallela* Loew, *A. pickeli* Lima, *A. serpentina* (Wied), *A. sororcula* Zucchi, *A. striata* Schiner, *A. suspensa* (Loew), *B. carambolae* Drew and Hancock, *B. dorsalis* (Hendel), *B. latifrons* (Hendel), *B. oleae* (Rossi), *C. capitata* (Wied.), *R. cerasi* (L.), *R. completa* Cresson, *R. cingulata* (Loew), *R. fausta* (Osten Sacken), *R. ferruginea* (Hendel), *R. mendax* (Curran), *R. meiiginii* (Loew), *R. pomonella* (Walsh), *R. suavis* (Loew), *R. tabellaria* (Fitch), *R. zephyria* Snow, and *Z. curcubitae* (Coquillett).

Information regarding the activities of parasitoids and predators on pestiferous fruit flies in the Americas is limited; therefore, this study aimed to compile a list of the diversity of parasitoid and predator species associated with tephritid fruit flies, as well as providing scientific evidence about the use of parasitoids and predators as biological control agents for fruit flies

## 2. Parasitoids 

The parasitoids that deposit an egg in the host are either solitary, more than one, or gregarious. When the parasitoids develop inside the host, they are endoparasitoids, and when development occurs externally, they are ectoparasitoids. Parasitoids can be either idiobionts or koinobionts. The former refers to those who kill their hosts shortly after oviposition, preventing further development, while the latter characterizes those who allow develop in the living hosts and kill them at the end of their cycle [28].

### 2.1. Native Parasitoids

In the Americas, there is a richness of 51 species of native parasitoids of fruit flies (Diptera, Tephritidae) of frugivorous fruit flies with economic importance included in 20 genera of 7 families (Braconidae, Diapriidae, Eurytomidae, Figitidae, Ichneumonidae, Mymaridae, and Pteromalidae). The Braconidae and Figitidae account for 64.7% of the fruit fly parasitoid species on the American continent. The family with the highest species richness is the Braconidae, with 47.1% of the total, followed by Figitidae with 17.6%, Pteromalidae with 17.6%, and Diapriidae with 11.8%. Ichneumonidae, Eurytomidae, and Mymaridae are represented by only one species each (5.9% one species each).

Most parasitoid species (53%) are associated with the genus *Anastrepha*, while 35.3%, 31.4%, and 7.8% of all parasitoid species are associated with *Rhagoletis*, *Ceratitis*, and *Bactrocera*, respectively. Table 1 shows the relationship between the parasitoid genera and the tephritid genera.

*Dicerataspis* Ashmead, *Eurytoma* Illiger *Lopheucoila* Weld, *Trichopria* Ashmead, *Tropideucoila* Ashmead, and *Rhoptromeris* Förster are associated exclusively with *Anastrepha*, while *Diachasma* Foerster, *Diachasmimorpha* Viereck, and *Eurytenes* Foerster are solely associated with *Rhagoletis*. The exotic genera of fruit flies, *Bactrocera* and *Ceratitis*, have no exclusive genus of parasitoid. *Doryctobracon* Enderlein, with a greater species richness, is associated with both *Anastrepha* and *Ceratitis*. *Coptera* Say and *Opius* Wesmael are the only genus of parasitoids associated with the 4 genera of tephritid fruit flies of economic importance.

Twenty host species associated with the Braconidae (*Anastrepha* (11 spp.); *Rhagolethis* (8 spp.); *Ceratitis* (1 sp.), were recorded. The Figitidae has 11 associated host species belonging to *Anastrepha* (7 spp.), *Rhagolethis* (3 spp.), and *Ceratitis* (1 sp.). The Diapriidae has 12 hosts species belonging to *Rhagoletis* (6 spp.), *Anastrepha* (5 spp.), and *Ceratitis* (1 sp.). The Pteromalidae has 6 hosts species *Anastrepha* (3 spp.), *Bactrocera* (2 spp.), *Ceratitis* (1 sp.). The Eurytomidae has 3 hosts (all *Anastrepha*). The Ichneumonidae has 2 hosts (all *Rhagoletis*), and Mymaridae with 1 (*Rhagoletis*) (Table 2). 

The native parasitoid species that has the largest number of hosts is *Doryctobacon areolatus* with 11 hosts (*Anastrepha* spp. and *Ceratitis capitata*), followed by *Utetes anastrephae* with 10 hosts (*Anastrepha* spp. and *Cerattis capitata*) (Table 2). *Anastrepha grandis* is known as the South American cucurbit fruit fly, a quarantine pest and the main fruit fly pest of cucurbitaceous plants that has no parasitoid associated. The absence of knowledge of natural enemies of *A. grandis* (parasitoids and predators) is probably due to the following factors: (1) Few studies carried out with this species when compared with other species of fruit flies of economic importance, (2) parasitoids have difficulty ovipositing in *A. grandis* larvae and eggs because the ovipositors of native parasitoid species may be unable to pierce the thick epicarp of the host fruits of this fly (Cucurbitaceae), (3) the absence of studies of pupae parasitodes and predators of *A. grandis*.

Parasitoids of economic fruit flies have been recorded in 20 of the 35 countries that are part of the Americas. Brazil has 19 native parasitoids and continental USA has 15 parasitoids, followed by Argentina and Mexico with 14 each (Table 3). The greatest occurrence of parasitoids in these countries is mainly due to the fact of a higher intensity of fruit fly parasitoid surveys compared to other countries. 

Most native parasitoids are solitary (81.1%), koinobionts (62.3%), or endoparasitoids (68.8%). The native species of Braconidae and Figitidae include endoparasitoid koinobionts that attack mainly larvae. The species of parasitoid idiobionts belong to the Diapriidae, Eurytomidae, and Pteromalidae. Ectoparasitism is found in the Eurytomidae and Pteromalidae. Most native parasitoids attack the larval stage (56.4%), pupa (32. 0%), and eggs (12.0%). There is no native parasitoid with a gregarious habit (Table 4).

### 2.2. Introduced Parasitoids

In the Americas and Hawaii, 29 species from 12 genera belonging to 7 families (Braconidae, Chalcididae, Diapriidae, Eulophidae, Figitidae, Ichneumonidae, and Pteromalidae) were introduced. Most introductions were species in the family Braconidae (72.4%). *Psyttalia* are associated with all fruit flies of economic importance, *Fopius*, *Dirhinus*, *Tetrastichus*, and *Pachycrepoideus* are associated with three fruit fly genera, *Coptera* and *Aceratoneuromyia* with two genera, while *Bathyplectes*, *Bracon*, and *Utetes* with only one genus (Table 5).

*Diachasmimorpha longicaudata* is the exotic parasitoid species with the most recorded hosts (12 species), and most widely distributed in the Americas. *Diachasmimorpha tryoni* and *Fopius arisanus* with 10 and 8 hosts, respectively, are also widely dispersed in the Americas (Table 6). The three braconid species previously mentioned are used against pestiferous tephritid species in Hawaii. Several introduced parasitoid species have been lab-reared, but until now they have not been released in the field, such as *Bracon celer* Szépligeti, *Fopius caudatus* (Szépligeti), *Fopius ceratitivorus* Wharton, *Psyttalia ponerophaga* (Silvestri), and *Utetes africanus* (Szépligeti), while other parasitoid species were released but did not establish, such as *Psyttalia concolor* (Szépligeti), *Psyttalia cyclogaster* (Thomson) (recorded previously as *Coeloreuteus formosanus*) (Watanabe) [69], *Psyttalia humilis* Silvestri, *Psyttalia perproxima* (Silvestri), and *Aganaspis daci* (Weld) (Table 6). Other parasitoid species were released but their establishment is unknown, either because there was no post-release follow-up, such as *Psyttalia rhagoleticola* (Sachtleben), *Coptera silvestrii* (Kieffer), and *Bathyplectes exiguus* Gravenhorst, or because of misidentifications with similar species that were collected from field surveys, such as *Fopius persulcatus* (Silvestri). Specimens belonging to *Fopius vandenboschi* (Fullaway) and *Fopius arisanus* (Sonan) recovered from field samples in Hawaii were mistakenly identified as *F. persulcatus* [115,116].

Most of the parasitoids introduced are solitary (89.6%), koinobiontes, (82.8%), and endoparasitoids (86.2%). Only 3 species present a gregarious habit and are restricted to the Eulophidae and Pteromalidae families. The larval stage is the most attacked stage (75.9%), followed by the pupa (13.8%) and egg (10.3%). Only *Dirhinus anthracia* Walker, *Dirhinus giffardii* Silvestri, and *Pachycrepoideus vindemmiae* (Rondani) are ectoparasitoids and idiobionts, while *Coptera silvestrii* is an endoparasitoid and idiobiont parasitoid (Table 7).

## 3. Predators

A total of 56 species of fruit fly predators occur in the Americas and Hawaii, including 11 spiders (Araneae: Araneidae, Lycosidae, Philodromidae, and Salticidae), 2 mites (Acari: Macrochelidae), 3 crickets (Orthoptera: Gryllidae and Trigonidiidae), 14 beetles (Coleoptera: Carabidae, Elateridae, and Stathyllinidae), 17 ants (Hymenoptera: Formicidae), 2 earwig (Dermaptera: Anisolabididae and Chelisochidae), 1 true bug (Reduviidae), 1 lizard (Reptilia: Polychrotidae), and 2 rodents (Rodentia: Vespertilionidae) (Table 8). From these predators, only the prey of spiders on the adult phase of fruit flies, the others prey on larvae and/pupae in the soil. 

The genus *Anastrepha* has 28 predators associated (9 with *A. ludens*, 8 with *A. suspensa*, 6 with *A. fraterculus*, and 5 with unidentified *Anastrepha* species), followed by *Rhagoletis* with 21 (16 species with *R. pomonella* and 5 with *R. mendax*), and *Ceratitis capitata* with 4. There are no predators of *Bactrocera* in the Americas.

Only 6 countries (Brazil, Canada, Cuba, Guatemala, Mexico, and USA) have a record of the economic importance fruit fly predators in the Americas. Canada has a richness of 20 species of predators, followed by USA (15 spp.), Brazil (11 spp.), Mexico (3 spp.), Cuba (1 sp.), and Guatemala (1 sp.) (Table 8).

Ants are a group of predatory insects that can be considered as pest control agents in some agroecosystems, regulating insect populations [198]. The predation of fruit flies by ants occurs when the larva leaves the fruit to bury itself in the soil for pupation [23]. Ants belonging to the genus *Pachycondyla*, *Pheidole*, *Pogonomyrmex*, and *Solenopsis* are important predators of *A. fraterculus* larvae in Brazil. *Solenopsis saevissima* was the most efficient species, with 42.86% of larvae removal in the field [25].

## 4. Biological Control Programs

Introduction of the invasive *Zeugodacus cucurbitae* (melon fly), *Bactocera dorsalis* (Hendel) (oriental fruit fly), and *Bactrocera latifrons* (Hendel) into Hawaii resulted initially in classical biological control programs, but later they became augmentative biological control programs [8,10,160,164]. Numerous parasitoid and predator species were introduced into Hawaii for classical biological control of *Bactrocera* spp. [69,178,179]. However, only the Asian-native larval parasitoid *Psyttalia fletcheri* (Silvestri) was successfully established in Hawaii on *Z. cucurbitae* with parasitism percentages that varied according to the host fruit species [8]. In an augmentative release program against the melon fly, *P. fletcheri* substantially reduced the number of emerged flies [155,199]. Other exotic parasitoid species, from Southern Asia and other regions, were successfully established on *B. dorsalis* in Hawaii, such as larval parasitoids *Diachasmimorpha longicaudata* (Ashmead), *Psyttalia incisi* (Silvestri), *Fopius vandenboschi* (Fullaway), and *Tetrastichus giffardianus* Silvestri, the egg–larval parasitoid *Fopius arisanus* (Sonan) and the pupal parasitoids *Dirhinus giffardii* (Silvestri) and *Pachycrepoideus vindemmiae* Rondani [154,200]. Augmentative releases of *D. longicaudata* made against *B. dorsalis* were inconsistent because they produced lower fly populations in the release plots one year and higher populations the next [10]. Although both *D. longicaudata* and *F. vandenboschi* were important biological control agents of *B. dorsalis*, *F. arisanus* has remained the most significant parasitoid of this tephritid species [8]. Because of the *F. arisanus* habit of attacking host eggs, which are more exposed below the fruit skin surface than larvae, this braconid parasitoid can achieve host parasitism percentages between 60% and 70% in the field [10]. In addition, *F. arisanus* was also the predominant species recovered from *B. latifrons* [158]. 

The establishment of the olive fruit fly *B. oleae* in California, USA, where it has spread to all commercial olive-producing areas since first being detected in 1998 [201], led to the development of a classical biological program control program in 2003 [167]. Several parasitoid species recovered from *B. oleae* collected from wild olives in Kenya, South Africa, Pakistan, or Namibia were imported to the USA. The introduced species were *Bracon celer* Szépligeti, *Psyttalia humilis* (Silvestri) (*P. humilis* from Kenya was previously referred to as *P.* cf. *concolor*) [165,202], *P. lounsburyi* (Silvestri), *P. ponerophaga* (Silvestri), and *Utetes africanus* (Silvestri) [117,118,165,168,169,170]. In addition, three exotic parasitoid species, i.e., the Australian-native *Fopius arisanus*, *Diachasmimorpha kraussii* (Fullaway), and *D. longicaudata*, coming from colonies in Hawaii, were also evaluated as potential biological control agents for *B. oleae* [122,203]. Although both *D. longicaudata* and *D. kraussii* were efficient against *B. oleae* [122], they were not considered for field releases because both braconid species are host-generalists. Given this, more specialized species such as the larval parasitoids *P. humilis* and *P. lounsburyi* were chosen to release in California [168,204]. Field release and recovery efforts were conducted from 2006 to 2013; both parasitoid species were recovered post-release, but only *P. lounsburyi* was established in California coastal regions [167,205]. Given these results, the parasitoid *P. lounsburyi* was mass-reared for release on a larger scale in olive-producing areas of California [206].

An augmentative biological control program against the introduced *B. carambolae*, the carambola fruit fly, was carried out in Northern Brazil (Amapá state) by releasing millions of *D. longicaudata* specimens [151]. Although *D. longicaudata* adapted to the Amazonian environment [17], it did not have a substantial effect in controlling the tephritid target. The Asian-native parasitoid *D. longicaudata* was previously introduced in 1994 into Brazil from Gainesville, Florida, United States, for use against *C. capitata* and *Anastrepha* spp. [17]. In 2012, a new biological control program against *B. carambolae* was started by introducing *F. arisanus* into Brazil from Hawaii; currently, this braconid parasitoid is reared on *C*. *capitata* eggs in different Embrapa laboratories and in the Moscamed Brazil facility [151].

The first classical biological control programs against *C. capitata* generally involved the introduction of parasitoid and predator species not only for the control of this pest but also for using them against other pestiferous tephritid species, such as *Bactrocera* spp. [21] or *Anastrepha* spp. [11]. The earliest classical biological control programs against *C. capitata* were carried out in Hawaii and date as far back as 1913. Because of these biocontrol programs, several parasitoid species, such as the larval parasitoids *Aceratoneuromyia indica* (Silvestri), *Diachasmimorpha fullawayi* (Silvestri), *D. tryoni* (Cameron), *D. longicaudata*, *F. vandenboschi*, *P. incisi*, *Aganaspis daci* (Weld), *Tetrastichus giffardianus* Silvestri, the egg–larval parasitoid *F. arisanus*, the pupal parasitoids *Coptera silvestrii* (Kieffer), *Dirhinus giffardii* Silvestri, and *P. vindemmiae*, were introduced to Hawaii and most of those parasitoid species were successfully established on *C. capitata* [120]. However, the Asian-native *F. arisanus*, since its establishment in the late 1940s, became the major parasitoid of *C. capitata* through substantial reductions in the medfly population in some habitats, apart from controlling *B. dorsalis* [8,115,207]. In spite of this, the implementation of classic biological control programs in Hawaii did not meet the objectives expected for the control of pestiferous fruit fly species, which motivated the development of mass-rearing of different parasitoid species for their periodic augmentative release in the field [208,209,210]. Therefore, using augmentative releases of parasitoids as a strategy into integrated management programs to control *C. capitata*, as well as for other pestiferous fruit fly species, has been encouraged since the 1980s [149,154,211]. Thus, augmentative releases of *D. tryoni* were performed in the late 1980s in Hawaii, due to its simplicity for mass-rearing and its host preference for *C. capitata* rather than *B*. *dorsalis* [149]. Those augmentative releases of *D. tryoni* were able to suppress medfly populations and the combination with sterile male fly releases had a greater effect on the pest [212]. However, high medfly populations still occur in Hawaii mainly in coffee plantations and at higher elevations [213]. Newly classical biological control programs carried out against *C. capitata* in Hawaii focused on the introduction of more specific parasitoid species [214,215]. New introduced species were the larval parasitoid *D. kraussii* and the Eastern African-native egg-pupal parasitoids *Fopius ceratitivorus* Wharton and *F. caudatus* (Szépligeti) [213]. Among these parasitoid species, *F. ceratitivorus* would be the most promising for improving overall suppression of medfly in Hawaii, due to its host specificity, lack of non-target impacts, and ability to complement *F. arisanus* [158,213,216,217]. 

Since the establishment of *C. capitata* in Hawaii, this pestiferous fruit fly species has been periodically introduced and erradicated in California, Florida, and Texas (USA) as well in Southern Mexico, although high medfly populations still remain throughout Central and South America [121]. The northward spread of *C. capitata* from Central America into Mexico, and also into the United States, has been constantly monitored along the Mexican/Guatemalan border by the international organization Mosca del Mediterráneo (MOSCAMED) (United States, Mexico, and Guatemala) [121]. Predominantly in this region, the vast areas cultivated with coffee, *Coffea arabica* L., which extend through the highlands of Guatemala, maintain high medfly populations. In addition to the use of the Sterile Insect Technique (SIT) to control medfly populations, augmentative parasitoid releases have also been carried out. *Diachasmimorpha tryoni* has been augmentatively released from the air into coffee cultivated areas affected by *C. capitata* in Guatemala over two years, which led to parasitism levels of up to 84% [146]. Previously, Cancino et al. [218] showed the significant effects of *D. longicaudata* mass-releases on *C. capitata* populations infesting coffee berries on the Mexico–Guatemala border. Similarly, augmentative releases of the Asian-native parasitoid *D. longicaudata* against medfly were carried out in Chiapas, Southern Mexico, during 2001 and 2002 on over 9000 ha of coffee plantations, reaching parasitism peaks of 61% and 69%, respectively [138]. In addition, augmentative releases of *D. krausii* and *F. arisanus*, either together or in combination with medfly sterile males, were made inside large field cages erected over coffee grown at different locations and altitudes in Guatemala. Results showed that the inclusion of both parasitoid species provided significant medfly suppression and the effect was frequently substantial [121]. These outcomes indicate that augmentative releases of parasitoids could be a complementary tool to control high medfly populations within an area-wide integrated fruit fly management (AWIFFM) approach [219]. The introduction from Kenya of the medfly-specific parasitoid *F. ceratitivorus* to Guatemala, and its establishment in the USDA-APHIS/MOSCAMED quarantine facility at San Miguel Petapa, points to a new process to strengthen the use of parasitoids against these medflies [156]. Considering differences in weather conditions and medfly density throughout the area of the Mexican/Guatemalan border, several parasitoid species with different bioecological features have been reared in Guatemala by the MOSCAMED Program [121]. Not all parasitoid species are equally effective under all likely conditions; preferences for temperature, moisture, and/or host density may vary [76,78]. 

Costa Rica was the first Central American country to develop a biological control program against *C. capitata* by introducing numerous parasitoid species mainly from Hawaii in the 1950s. Thus, *F. arisanus*, *D. longicaudata*, *A. indica*, *P. concolor*, and *P. vindemmiae* were released and recovered in Costa Rica, but the impact on *C. capitata* was not significant [11]. In the 1980s, a classic biological control program facilitated the introduction to Costa Rica of four parasitoid species from Africa, i.e., *Diachasmimorpha fullawayi*, *Psyttalia perproxima* (Silvestri) (recorded as *P. perproximus* (Silvestri)) [119], *Fopius caudatus*, and *F. silvestrii* (Wharton); this last species was previously misidentified as *F. caudatus* [220]. The four braconid parasitoid species were directly released in the field but there was no recovery of them post-release [119]. Currently, both *D. longicaudata* and *P. vindemmiae* are mass-reared in Costa Rica for fruit fly biological control, although little information is available on their present status parasitizing *C. capitata* [133]. The fruit fly biological control program developed by Costa Rica in the 1950s was essential for promoting the use of parasitoids in other Latin American countries affected by the medfly. Thus, *D. longicaudata*, *A. indica*, and *P. vindemmiae* were mainly provided by Costa Rica to Nicaragua, Panamá, El Salvador, Guatemala, Trinidad and Tobago (Central America), Argentina, Bolivia, Colombia, Perú, and Venezuela (South America).

The larval parasitoid *T. giffardianus* was the first exotic species introduced into both Brazil and Argentina during the 30s and 40s, respectively, for *C. capitata* control. Low numbers of individuals were released in both countries. In Brazil, this eulophid parasitoid was recovered from medfly puparia after 60 years from its first release [17], but in Argentina there has been no evidence of its permanent establishment at any release site [16]. New biological control programs that involved the introduction of several exotic parasitoid species into Argentina from Mexico and Costa Rica were carried out between the 1960s and 1990s. The establishment on *C. capitata* of three released parasitoid species, *D. longicaudata*, *A. indica*, and *P. vindemmiae*, was verified in Argentina, although without exercising significant control on this tephritid pest [16]. However, open-field augmentative releases of *D. longicaudata* mass-reared on irradiated larvae of the tsl Vienna-8 medfly strain (named as “*D. longicaudata* tsl-Cc line”) have recently been carried out in fruit-growing areas of Central–Western Argentina. Post-release data showed up to 75% of *C. capitata* mortality due to the *D. longicaudata* releases in fig crops [126,127]. Later, augmentative releases of the *D. longicaudata* tsl-Cc line were carried out to assess the effectiveness of parasitoid females in killing medfly larvae infesting peach and orange inside a field cage in the subtropical environment of the northwestern Argentina. Parasitoid effectiveness reached up to 50% in infested peaches [128]. Recent studies on the mass-rearing of the neotropical pupal parasitoid *Coptera haywardii* (Ogloblin) using gamma-irradiated larvae of the tsl Vienna-8 medfly strain as the host have been carried out at the BioPlanta San Juan facility [20]. Furthermore, the possibility of augmentative releases of *C. haywardii* for medfly control is currently being evaluated in Argentina. Augmentative releases of *D. longicaudata* against *C. capitata* were also carried out in different Brazilian regions, but recoveries of this braconid parasitoid were more associated with *Anastrepha* spp. [17,131,132]. This braconid parasitoid species has been previously introduced in 1994 into Brazil from Gainesville, Florida, United States, for use against *C. capitata* and *Anastrepha* spp. [151].

Historically, the introduction and release of exotic hymenopterous parasitoid species for biological control of native pestiferous *Anastrepha* species have been mainly standardized from the 1930s in Puerto Rico, Costa Rica, and Mexico, as well as in many Latin American countries. Most of these parasitoid species had first been introduced into Hawaii including *D. longicaudata*, *D. fullawayi*, *D. tryoni*, *P. incisi*, *P. concolor*, *P. fletcheri*, *F. arisanus*, *F. vandenboschi*, *A. daci*, *T. giffardianus*, *D. giffardii*, *A. indica,* and *P. vindemmiae*. A few species were able to establish in the released areas and they were able to control the target *Anastrepha* species. The status of all those introduced parasitoid species was discussed by Ovruski et al. [11].

Since 1992, Mexico has carried out the main biological control program against pestiferous *Anastrepha* species in the Americas. This national program, sponsored by the Mexican government, focuses on achieving free and/or low-prevalence areas of four economical and quarantined *Anastrepha* species, i.e., *A. ludens*, *A. obliqua*, *A. serpentina*, and *A. striata* [14]. Therefore, three exotic braconid parasitoid species, namely *D. longicaudata*, *D. tryoni* [221], and *F. arisanus* [152] have been mass-reared at the Moscafrut facility in Metapa de Dominguez, Chiapas, Mexico and released. Of the three parasitoids species, 50 million pupae parasitized by *D. longicaudata* were produced weekly [222]. The first parasitoid augmentative releases in Mexico started in the late 80s, when an average of 1500 *D. longicaudata* parasitoids per ha. was released on almost 200 ha in the Valle of Mazapa de Madero, Chiapas, Mexico. Significant reductions in *Anastrepha* species were recorded with an average of 60% parasitism. This test was performed in a diverse range of *Anastrepha* species [223]. This was an important step towards implementing an IPM program in *Anastrepha* populations in Mexico. Consequently, new augmentative area-wide releases of the *D. longicaudata*, mass-reared, have been carried out by air or from the ground in different Mexican states [14,15,222]. Releases of *D. longicaudata* have been continuously focused on wild areas and backyard orchards to prevent fruit fly dispersion into commercial crops, which has caused substantial reductions in numbers of *Anastrepha* adults [56,137,218,224,225]. However, despite the good outcomes achieved with *D. longicaudata*, the Mexican National Program against *Anastrepha* spp. fruit flies has turned attention to the many neotropical-native parasitoid candidates for augmentative release with the chance of strategically increasing the mortality inflicted on pestiferous *Anastrepha* species. Thus, native parasitoid species better adapted to certain natural environmental conditions to low host densities, or that may attack other developmental stages of the pest, can be used to complement exotic parasitoid species [146,226,227,228]. Therefore, colonization of several neotropical parasitoid species and the mass-rearing of some of the species took place in Mexico [18,57,229]. Thus, the native *C. haywardii* is currently a suitable candidate for use with *D. longicaudata* in augmentative area-wide releases against *Anastrepha* spp. in Mexico [19,230,231].

The introduction of *A. suspensa* into Florida in 1965 led to the establishment of a biological control program which was developed by importing at least 11 parasitoid species from Hawaii, France, and Latin America between the early and late 1970s. Among introduced parasitoid species both *D. longicaudata* and the neotropical-native *Doryctobracon areolatus* (Szépligeti) has been successfully established into Florida [11]. Populations of *A. suspensa* decreased by 40% in some areas in the years following releases of the two braconid parasitoid species, but *A. suspensa* continued as a serious pest in Florida [159]. In view of this, a *D. longicaudata* mass-rearing and augmentative releases were carried out later, which generated significant reductions in *A. suspensa* populations in urban and suburban areas of Florida [144,232]. Because of these releases *D longicaudata* replaced *D. areolatus* as the major parasitoid of *A. suspensa* in the southern portion of Florida, while *D. areolatus* predominated in the northern sector [46,233].

In Brazil, the exotic *D. longicaudata* has shown the ability to adapt and settle in different environments, either in semi-arid or tropical areas, to control pestiferous *Anastrepha* species [17,130,131,132,151]. Augmentative releases of *D. longicaudata* against *A. fraterculus* in wild vegetation areas near citrus crops in the State of São Paulo, Brazil caused a reduction of 30% in adult fly numbers [17]. Experimental studies under laboratory conditions showed the ability of *F. arisanus* to develop successfully in *A. fraterculus* eggs compared with *C. capitata* eggs [234]. This trait makes *F. arisanus* a complementary alternative to the use of *D. longicaudata* against *A. fraterculus*. 

Parasitoid species introduced into Argentina for the biological control of *C. capitata* were also released against *A. fraterculus*. Three exotic species, *D. longicaudata*, *A. indica*, and *P. vindemmiae* were established after releases in different Argentinean fruit-growing areas [16]. In addition to the introduced parasitoid species, both Brazil and Argentina followed the initiative of the Mexican National Fruit Fly Program regarding the use of neotropical parasitoids for *Anastrepha* control. Thus, several native parasitoid species were colonized and lab-reared to be evaluated as biocontrol agents of *A. fraterculus* in Brazil, i.e., *Aganaspis pelleranoi* [235,236] and *Doryctobracon brasiliensis* [237,238], and in Argentina, *C. haywardii* [20,97,239], *A. pelleranoi* [240,241], *Doryctobracon crawfordi* [240], and *Opius bellus* [74,240].

In Peru, near to the border with Chile in the Department of Tacna, the National Medfly Program was releasing the parasitoid *D. longicaudata* weekly (reared in La Molina Facility in Lima) in the late 90s in order to reduce *C. capitata* populations in olive orchards. More than 50% parasitism was achieved during the two years with the periodical massive releases of *D. longicaudata*. The market quality of olives was maintained at a high level with the implementation of an of IPM program against *C. capitata* in which the biological control played an important role.

There is only one document that records the introduction and releases of parasitoid species during the 1950s into California against *R. completa*, *R. indifferens*, and *R. fausta* [161], but the results of these releases were unsatisfactory [242]. Much later, evaluations of the parasitism capability of *Psyttalia humilis* (referred to as *P.* cf. *concolor*) on *R. completa* were carried out only under laboratoty conditions; this parasitoid was introduced into California for *B. oleae* biological control [202].

In the Western United States there are two parasitoids, *Opius lectoides* Gahan and *O. downesi*, which generally attack *Rhagoletis zephyria* Snow and *R. pomonella* in Oregon. About 60% of pupae were parasitized by these parasitoids on native host plants, while less than 2% on apple were attacked. Both species of *Opius* have short ovipositors, which may not be long enough to reach host larvae in the larger apple fruits. Alternatively, *Diachasma allaeum* has a much longer ovipositor and has been very successful in parasitizing larvae in apples in the Eastern United States [243].

One of the major concerns in the use of predators in pest control is intraguild predation [24]. Intraguild predation occurs among natural enemies in biological control systems, where one natural enemy (the intraguild predator) attacks another species of natural enemy (the intraguild prey), whereas they also compete for the same pest [27]. There are two types of intraguild predation (PGI) between predators and parasitoids: (1) The predator can directly predate the parasitoid, feeding from the immature phase then externally to the host and adult phase; (2) the predator can predate the parasitic host, directly consuming the host and, indirectly, the larva of the parasitoid [244]. The effect of the presence of intraguild predators on the intraguild prey was often negative, but sometimes no significant effect was detected [27]. Although predators are not the focus of fruit fly biological control programs, they have a very important role in conservation biological control, and it is necessary to intensify studies that evaluate or use agricultural techniques that do not affect an assemblage of predators of fruit flies, such as the use of selective pesticides.

## Figures and Tables

**Table 1 insects-11-00662-t001:** Association between the genus of native parasitoids (Hymenoptera) and the genus of fruit flies with economic importance in the Americas.

Parasitoid Species	Genus of Tephritidae			
	*Anastrepha*	*Bactrocera*	*Ceratitis*	*Rhagoletis*
**Braconidae**				
*Asobara*	X		x	
*Diachasma*				x
*Diachasmimorpha*				X
*Doryctobracon*	X		x	
*Eurytenes*				x
*Opius*	X	x	x	x
*Utetes*	X			x
**Diapriidae**				
*Coptera*	X	x	x	x
*Trichopria*	X			
**Eurytomidae**				
*Eurytoma*	X			
**Figitidae**				
*Aganaspis*	X		x	x
*Dicerataspis*	X			
*Lopheucoila*	X			
*Odontosema*	X		x	
*Tropideucoila*	X			
*Rhoptromeris*	X			
**Ichneumonidae**				x
*Phygadeuon*				
**Mymaridae**				
*Anaphes*				x
**Pteromalidae**				
*Pteromalus*		x	x	
*Spalangia*		x	x	

**Table 2 insects-11-00662-t002:** Native parasitoid species and their fruit fly hosts in the Americas.

Parasitoid Species	Host Fruit Fly Species	References
**Braconidae**		
*Asobara anastrephae*	*A. fraterculus*, *A. obliqua*, *A. striata*, *C. capitata*	[17,29,30,31,32,33,34,35,36]
*Diachasma alloeum*	*R. pomonella*, *R. mendax*, *R. zephyria*	[37]
*Diachasma ferrugineum*	*R. pomonella*	[38]
*Diachasmimorpha martinalujai*	*R. pomonella*	[39]
*Diachasmimorpha mellea*	*R. cingulata*, *R. mendax*, *R. pomonella*, *R. zephyria*	[37,39,40]
*Diachasmimorpha sublaevis*	*R. completa*	[41]
*Doryctobracon adaimei*	*A. fraterculus*, *A. striata*	[35,42]
*Doryctobracon areolatus*	*A. distincta*, *A. fraterculus*, *A. ludens*, *A. obliqua*, *A. pseudoparallela*, *A. pickeli*, *A. serpentina*, *A. sororcula*, *A. striata*, *A. suspensa*, *C. capitata*	[17,29,30,43,44,45,46,47,48,49,50,51,52,53,54,55,56,57,58,59,60,61,62]
*Doryctobracon auripennis*	*A. serpentina*	[55]
*Doryctobracon brasiliensis*	*A. fraterculus*, *A. obliqua*, *A. pickeli*, *A. serpentina*, *A. sororcula*	[17,43,44,48,59,61,63,64,65]
*Doryctobracon capsicola*	*A. serpentina*	[55]
*Doryctobracon crawfordi*	*A. distincta*, *A. fraterculus*, *A. ludens*, *A. obliqua*, *A. serpentina*, *A. sororcula*, *A. striata*, *C. capitata*	[17,29,47,49,50,52,53,55,58,59,64,66,67,68]
*Doryctobracon fluminensis*	*A. fraterculus*, *A. pickeli*, *A. pseudoparallela*	[17,43,44,50]
*Doryctobracon toxotrypanae*	*A. curcuvicauda*	[69]
*Doryctobracon trinidadensis*	*A. serpentina*, *A. striata*	[29,70]
*Doryctobracon zeteki*	*A. serpentina*, *A. striata*	[29,45,50,55]
*Eurytenes (Stigmatopoea) maya*	*R. pomonella*	[39]
*Opius bellus*	*A. distincta*, *A. fraterculus*, *A. obliqua*, *A. pickeli*, *A. serpentina*, *A. sororocula*, *A. striata*, *C. capitata*, *R. ferrugínea*	[17,29,30,31,32,43,44,48,54,59,60,63,65,71,72,73,74,75]
*Opius downesi*	*R. pomonella*, *R. tabellaria*, *R. zephyria*	[37]
*Opius hirtus*	*A. curvicauda*, *A. ludens*, *A. obliqua*	[18,76,77,78,79]
*Utetes anastrephae*	*A. distincta*, *A. fraterculus*, *A. ludens*, *A. obliqua*, *A. pickeli*, *A. serpentina*, *A. sorocula*, *A. striata*, *A. suspensa*, *C. capitata*	[17,18,30,43,44,46,47,48,50,52,56,59,60,61,62,63,66,75,80,81,82,83,84,85,86,87]
*Utetes canaliculatus*	*R. pomonella*	[37,69,88,89]
*Utetes lectoides*	*R. pomonella*	[37]
*Utetes richmondi*	*R. carnivora*, *R. mendax*	[37,69]
*Utetes tomoplagiae* (as *Opius tomoplagiae*)	*A. fraterculus*	[17,90,91,92]
**Diapriidae**		
*Coptera cingulatae*	*R. cingulata*, *R. fausta*, *R. pomonella*, *R. suavis*	[69]
*Coptera evansi*	*R. completa*, *R. fausta*	[41,69]
*Coptera haywardi*	*A. fraterculus*, *A. ludens*, *A. obliqua*, *A. serpentina*, *A. sororcula*, *C. capitata*	[16,17,18,52,93,94,95,96,97,98]
*Coptera pomonellae*	*R. pomonella*, *R. suavis*	[69]
*Coptera occidentalis*	*R. cingulata*, *R. cerasi*, *R. completa*	[41,69]
*Trichopria anastrephae*	*A. fraterculus*, *A. obliqua*, *A. serpentina*, *C. capitata*	[17,48,98,99,100]
**Eurytomidae**		
*Eurytoma sivinskii*	*A. ludens*, *A. obliqua*, *A. serpentina*	[101]
**Figitidae**		
*Aganaspis alujai*	*R. completa*, *R. ramosae*, *R. zoqui*	[102,103]
*Aganaspis nordlanderi*	*A. fraterculus*, *A. striata*, *C. capitata*	[17,104,105]
*Aganaspis pelleranoi*	*A. distincta*, *A. fraterculus*, *A. ludens*, *A. obliqua*, *A serpentina*, *A. striata*, *C. capitata*	[17,18,44,50,52,56,59,61,66,104,105,106]
*Dicerataspis grenadensis*	*A. fraterculus*	[17,107]
*Lopheucoila anastrephae*	*A. fraterculus*, *A. pseudoparallela*	[17,34,59,105]
*Odontosema anastrephae*	*A. fraterculus*, *A. ludens*, *A. obliqua*, *C. capitata*	[17,34,52,56,59,66,75,104,105,108,109]
*Odontosema albinerve*	*A. fraterculus. A. serpentina*, *C. capitata*	[48,110]
*Tropideucoila weldi*	*A. fraterculus*, *A. sororcula*	[111]
*Rhoptromeris haywardi*	*A. fraterculus*, *C. capitata*	[16]
**Ichneumonidae**		
*Phygadeuon wiesmanni*	*R. pomonella*, *R. cerasi*	[112]
**Mymaridae**		
*Anaphes conotracheli*	*R. pomonella*	[112]
**Pteromalidae**		
*Pteromalus kapaunae*	*B. oleae*	[69,113]
*Spalangia cameroni*	*Z. curcubitae*	[69]
*Spalangia endius*	*A. fraterculus*, *A. obliqua. C. capitata*	[98]
*Spalangia gemina*	*A. fraterculus*, *A. obliqua*, *C. capitata*	[93,98,114]
*Spalangia impunctata*	*C. capitata*	[98]
*Spalangia leiopleura*	*C. capitata*	[98]
*Spalangia nigra*	*Z. cucurbitae*	[69]
*Spalangia simplex*	*A. serpentina*, *C. capitata*	[98]

**Table 3 insects-11-00662-t003:** List and distribution of native Hymenopteran parasitoid on fruit-infesting Tephritidae in the Americas.

Native Parasitoid Species	Countries
**Braconidae**	
*Asobara anastrephae*	ARG, BRA, COL, PAN
*Diachasma alloeum*	USA
*D. ferrugineum*	USA
*Diachasmimorpha martinalujai*	MEX
*D. sublaevis*	USA
*D. mellea*	MEX, USA
*Doryctobracon adaimei*	BRA
*D. areolatus*	ARG, BOL, BRA, COL, COR, ECU, ELS, FRG, GUA, MEX, PAN, USA, VEN
*D. auripennis*	PAN
*D. brasiliensis*	ARG, BOL, BRA
*D. capsicola*	PAN
*D. crawfordi*	ARG, BOL, BRA, COL, COR, ECU, ELS, GUA, MEX, PAN, VEN
*D. fluminensis*	BRA, VEN
*D. toxotrypanae*	COR, ELS
	MEX, VEN
*D. trinidadensis*	PAN, TRT
*D. zeteki*	COL, PAN, VEN
*Eurytenes (Stigmatopoea) maya*	MEX
*Opius bellus*	ARG. BEL, BOL. BRA, COR, FRG, MEX, PAN, VEN
*O. downesi*	CAN, USA
*O. hirtus*	COR, DOR, MEX,
*Utetes anastrephae*	ARG, BOL, BRA, COL, COR, CUB, ECU, ELS, FRG, GUA, MEX, PUR, USA, VEN
*U. canaliculatus*	CAN, USA, MEX
*U. richmondi*	CAN, USA
*U. lectoides*	CAN, USA
**Diapriidae**	
*Coptera cingulatae*	USA
*C. evansi*	USA
*C. haywardi*	ARG. BRA. MEX, VEN
*C. pomonellae*	USA
*C. occidentalis*	USA
*Trichopria anastrephae*	ARG, BRA, VEN
**Eurytomidae**	
*Eurytoma sivinskii*	MEX
**Figitidae**	
*Aganaspis alujai*	MEX, USA
*A. nordlanderi*	ARG, BRA, COR
*A. pelleranoi*	ARG, BOL, BRA, COL, COR, ECU, FRG, GUA, MEX, PER, VEM
*Dicerataspis grenadensis*	ARG
*Lopheucoila anastrephae*	ARG, BOL, BRA, PAN
*Odontosema anastrephae*	ARG, BOL, BRA, COL, COR, MEX
*O. albinerve*	BRA
*Tropideucoila weldi*	BRA
*Rhoptromeris haywardi*	ARG
**Ichneumonidae**	
*Phygadeuon wiesmanni*	CAN
**Mymaridae**	
*Anaphes conotracheli*	CAN
**Pteromalidae**	
*Pteromalus kapaunae*	USA
*Spalangia cameroni*	USA
*S. endius*	USA, BRA
*S. gemina*	BRA
*S. impunctata*	BRA
*S. leiopleura*	BRA
*S. nigra*	USA
*S. simplex*	BRA

Countries: ARG, Argentina; BEL, Belize; BOL, Bolivia; BRA, Brazil; CAN, Canada; COL, Colombia; COR, Costa Rica; CUB, Cuba; DOR, Dominican Republica; ECU, Ecuador; ELS, El Salvador; FRG, French Guiana; GUA, Guatemala; MEX, Mexico; PAN, Panama; PER, Peru; PUR, Puerto Rico, TRT, Trinidad and Tobago; VEN, Venezuela; USA, United States of America.

**Table 4 insects-11-00662-t004:** Guilds of native parasitoids of tephritid fruit flies with economic importance in the Americas.

Native Parasitoid Species	Parasitism Modes	Feeding Types	Host Stage Attacked
**Braconidae**			
*Asobara anastrephae*	S, K	En	L3
*Diachasma alloeum*	U	En	L
*D. ferrugineum*	U	U	L
*Diachasmimorpha martinalujai*	S, K	En	L
*D. mellea*	S, K	En	L
*D. sublaevis*	S, K	En	L
*Doryctobracon adaimei*	S, K	En	L
*D. areolatus*	S, K	En	E-L1
*D. auripennis*	S, K	En	L
*D. brasiliensis*	S, K	En	L3
*D. capsicola*	S, K	En	L
*D. crawfordi*	S, K	En	L3
*D. fluminensis*	S, K	En	L
*D. toxotrypanae*	S, K	En	L
*D. trinidadensis*	S, K	En	L
*D. zeteki*	S, K	En	L
*Eurytenes (Stigmatopoea) maya*	U	U	L
*Opius bellus*	S, K	En	L2-L3
*O. downesi*	S, K	En	L
*O. hirtus*	S, K	En	L
*Utetes anastrephae*	S, K	En	L3
*U. canaliculatus*	U	En	E
*U. richmondi*	U	En	E
*U. lectoides*	U	U	E
**Diapriidae**			
*Coptera cingulatae*	S, I	En	P
*C. evansi*	S, I	En	P
*C. haywardi*	S, I	En	P
*C. pomonellae*	S, I	En	P
*C. occidentalis*	S, I	En	P
*Trichopria anastrephae*	S, I	En	P
**Eurytomidae**			
*Eurytoma sivinskii*	S, I	Ec	P
**Figitidae**			
*Aganaspis alujai*	S, K	En	L3
*A. pelleranoi*	S, K	En	L3
*A. nordlanderi*	S, K	En	L3
*Dicerataspis flavipes*	S, K	En	L3
*D. grenadensis*	S, K	En	L3
*Lopheucoila anastrephae*	S, K	En	L3
*Odontosema anastrephae*	S, K	En	L3
*O. albinerve*	S, K	En	L3
*Tropideucoila weldi*	S, K	En	L
*Rhoptromeris haywardi*	S, K	En	L
**Ichneumonidae**			
*Phygadeuon wiesmanni*	U	U	P
**Mymaridae**			
*Anaphes conotracheli*	U	U	E
**Pteromalidae**			
*Pteromalus kapaunae*	S, I	Ec	P
*Spalangia cameroni*	S, I	Ec	P
*S. endius*	S, I	Ec	P
*S. gemina*	S, I	Ec	P
*S. impunctate*	S, I	Ec	P
*S. leiopleura*	S, I	Ec	P
*S. nigra*	S, I	Ec	P
*S. simplex*	S, I	Ec	P

S, Solitary; G, Gregarious; I, Idiobiont; K, Koinobiont; Ec, Ectoparasitoid; En, Endoparasitoid; L, Larvae; L1, Larva fisrt instar; L2, Larva second instar; L3, Larva third instar; P, Pupae; U, Unknown.

**Table 5 insects-11-00662-t005:** Association between genera of introduced parasitoids (Hymenoptera) with the genus of fruit flies with economic importance in the Americas and Hawaii.

Parasitoid Species	*Anastrepha*	*Bactrocera*	*Ceratitis*	*Rhagoletis*	*Zeugodacus*
**Braconidae**					
*Bracon*		X			
*Diachasmimorpha*	**X**	**X**	**X**		
*Fopius*	X	X	X		
*Psyttalia*	**X**	**X**	**X**	**X**	**X**
*Utetes*		**X**			
**Chalcididae**					
*Dirhinus*	X	X	X		
**Diapriidae**					
*Coptera*		X	**X**		
**Eulophidae**					
*Aceratoneuromyia*	**X**		**X**		
*Tetrastichus*	X	X	X		
**Figitidae**					
*Aganaspis*	**X**		**X**		
**Ichneumonidae**					
*Bathyplectes*				X	
**Pteromalidae**					
*Pachycrepoideus*	**X**	X	**X**		

**Table 6 insects-11-00662-t006:** Introduced parasitoid species and their target tephritid fruit fly species in the Americas and Hawaii.

Introduced Parasitoid Species	Parasitoid Status				Importing Country	Fruit Fly Species	Reference
	Lab-Reared	Field-Cage Released	Open-Field Released	Established			
**Braconidae**							
*Bracon celer*	Yes	No	No	No	USA	*B. oleae*	[117,118]
*Diachasmimorpha fullawayi*	No	No	Yes	?	COR	*C. capitata*	[119]
	No	No	Yes	?	PUR	*A. obliqua A. suspensa*	[81]
	No	No	Yes	?	USA (H)	*Z. cucurbitae*	[69]
	No	No	Yes	Yes	USA (H)	*C. capitata*	[120]
*Diachasmimorpha krausii*	Yes	Yes	No	No	GUA	*C. capitata*	[121]
	Yes	No	No	No	USA	*B. oleae*	[122]
	Yes	No	Yes	Yes	USA (H)	*B. latifrons*	[123]
*Diachasmimorpha longicaudata*	Yes	Yes	Yes	Yes	ARG	*A. fraterculus C. capitata*	[124,125,126,127,128]
	No	No	Yes	?	BOL	*C. capitata*	[59,129]
	Yes	No	Yes	Yes	BRA	*A. fraterculus C. capitata*	[17,130,131,132]
	Yes	No	Yes	Yes	COR	*C. capitata*	[75,133]
	Yes	No	Yes	Yes	ELS	*C. capitata*	[134]
	Yes	No	Yes	Yes	FRG	*A. obliqua A. striata*	[135]
	Yes	No	Yes	Yes	GUA	*C. capitata*	[47]
	Yes	Yes	Yes	Yes	MEX	*A. curvicauda A. fraterculus A. ludens A. obliqua A. serpentina* *A. striata C. capitata*	[15,19,52,56,136,137,138]
	No	No	Yes	Yes	NIC	*A. obliqua* *C. capitata*	[139,140]
	No	No	Yes	Yes	PAN	*A. ludens* *C. capitata*	[29,75]
	Yes	No	Yes	Yes	PER	*C. capitata*	[141,142]
	No	No	Yes	Yes	TRT	*C. capitata*	[143]
	Yes	No	Yes	Yes	USA	*A. suspensa*	[46,144]
	Yes	No	Yes	Yes	USA (H)	*Z. cucurbitae B. dorsalis* *B. latifrons C. capitata*	[8,21,69]
	No	No	Yes	?	VEN	*C. capitata*	[143]
*Diachasmimorpha tryoni*	Yes	Yes	No	No	ARG	*C. capitata*	[145]
	No	No	No	?	BRA	*C. capitata*	[120]
					COR	*C. capitata*	[70,75]
	Yes	No	Yes	Yes	GUA	*C. capitata*	[146]
	Yes	Yes	Yes	Yes	MEX	*C. capitata*	[147]
	No	No	Yes	?	PUR	*A. obliqua A. suspensa*	[81]
	Yes	No	Yes	No	USA	*R. completa*	[41]
	Yes	No	Yes	?	USA	*A. suspensa*	[148]
	Yes	No	Yes	Yes	USA (H)	*Z. cucurbitae C. capitata*	[69,149,150]
*Fopius arisanus*	No	No	Yes	No	ARG	*C. capitata*	[16]
	Yes	No	No	No	BRA	*C. capitata*	[151]
	Yes	No	Yes	Yes	COR	*C. capitata*	[75]
	Yes	Yes	No	No	GUA	*C. capitata*	[121]
	Yes	No	Yes	No	MEX	*A. ludens A. obliqua A. serpentina* *A. striata* *C. capitata*	[148,152,153]
	No	No	?	?	PER	*C. capitata*	[141,142]
	Yes	Yes	Yes	Yes	USA (H)	*Z. cucurbitae B. dorsalis B. latifrons C. capitata*	[21,154,155]
*Fopius caudatus*	No	No	?	?	COR	*C. capitata*	[119]
	Yes	No	No	No	GUA	*C. capitata*	[156]
	Yes	No	No	No	USA (H)	*B. latifrons C. capitata*	[157]
*Fopius ceratitivorus*	Yes	No	No	No	GUA	*C. capitata*	[156]
	Yes	No	No	No	USA (H)	*C. capitata*	[21,158]
*Fopius persulcatus*	No	No	No	No	USA	*A. suspensa*	[159]
	Yes	No	Yes	?	USA (H)	*B. dorsalis B. latifrons C. capitata*	[115,160]
*Fopius silvestrii*	Yes	No	No	No	COR	*C. capitata*	[119,161]
*Fopius vandenboschi*	No	No	Yes	?	COR	*C. capitata*	[75]
	No	No	Yes	?	MEX	*A. ludens A. obliqua*	[162]
	Yes	No	Yes	Yes	USA (H)	*B. dorsalis*	[21,69]
*Psyttalia concolor*	No	No	Yes	?	BOL	*C. capitata*	[143]
	No	No	Yes	?	COL	*C. capitata*	[66,139]
	No	No	Yes	No	COR	*C. capitata*	[162,163]
	No	No	Yes	?	ELS	*C. capitata*	[134]
	No	No	Yes	?	GUA	*C. capitata*	[143]
	No	No	Yes	No	PAN	*C. capitata*	[75]
	No	No	Yes	?	PER	*C. capitata*	[141]
	No	No	Yes	No	PUR	*A. obliqua A. suspensa*	[81,162]
	No	No	Yes	No	USA	*A. suspensa*	[148]
*Psyttalia cyclogaster*	No	No	Yes	No	USA	*Z. cucurbitae*	[69]
*Psyttalia fletcheri*	No	No	?	?	BRA	*C. capitata*	[120]
	No	No	Yes	No	PUR	*A. obliqua A. suspensa*	[81,162]
	No	No	Yes	No	USA	*A. suspensa*	[148]
	Yes	Yes	Yes	Yes	USA (H)	*Z. cucurbitae C. capitata*	[8,21,155]
*Psyttalia humilis*	Yes	No	Yes	No	USA (H)	*Z. cucurbitae*	[69,164,165]
	Yes	No	Yes	No	USA	*B. oleae*	[166,167]
*Psyttalia incisi*	No	No	Yes	No	COR	*C. capitata*	[162,163]
	No	No	Yes	No	MEX	*A. ludens A. obliqua*	[148,162]
	Yes	No	Yes	Yes	USA (H)	*B. dorsalis B. latifrons*	[150]
*Psyttalia lounsburyi*	Yes	No	Yes	Yes	USA	*B. oleae*	[167,168]
*Psyttalia perproxima*	No	No	Yes	No	COR	*C. capitata*	[119]
*Psyttalia ponerophaga*	Yes	No	No	No	USA	*B. oleae*	[169]
*Psyttalia rhagoleticola*	No	No	Yes	?	CAN	*R. pomonella*	[112]
*Utetes africanus*	Yes	No	No	No	USA	*B. oleae*	[167,170]
**Chalcididae**							
*Dirhinus anthracia*	Yes	No	Yes	Yes	USA (H)	*Z. cucurbitae*	[69,150]
*Dirhinus giffardii*	No	No	Yes	?	BOL	*C. capitata*	[171]
	No	No	Yes	?	COL	*C. capitata*	[139]
	No	No	Yes	?	COR	*C. capitata*	[163]
	No	No	Yes	?	MEX	*A. ludens A. obliqua*	[148]
	No	No	Yes	?	PER	*C. capitata*	[141]
	No	No	Yes	No	USA	*A. suspensa*	[148]
	Yes	No	Yes	Yes	USA (H)	*Z. cucurbitae C. capitata*	[69,172]
**Diapriidae**							
*Coptera silvestrii*	Yes	No	Yes	?	USA (H)	*Z. cucurbitae C. capitata*	[69,120]
**Eulophidae**							
*Aceratoneuromyia indica*	Yes	No	Yes	Yes	ARG	*A. fraterculus C. capitata*	[86]
	No	No	Yes	Yes	BOL	*C. capitata*	[59,139]
	No	No	Yes	Yes	COL	*A. fraterculus A. striata C. capitata*	[66,139]
	No	No	Yes	?	COR	*C. capitata*	[75]
	No	No	Yes	?	GUA	*C. capitata*	[173]
	No	No	Yes	Yes	MEX	*A. ludens A. obliqua*	[52,174]
	No	No	Yes	Yes	NIC	*C. capitata*	[75]
	No	No	Yes	?	PAN	*C. capitata*	[75]
	No	No	?	?	PER	*C. capitata*	[141]
	No	No	?	?	TRT	*C. capitata*	[143]
	No	No	Yes	Yes	USA (H)	*C. capitata*	[120]
	No	No	Yes	Yes	VEN	*C. capitata*	[143]
*Tetrastichus giffardianus*	No	No	Yes	No	ARG	*C. capitata*	[16]
	Yes	No	Yes	Yes	BRA	*C. capitata*	[120,175]
	No	No	Yes	?	PER	*C. capitata*	[141]
	No	No	Yes	No	PUR	*A. obliqua A. suspensa*	[81]
	No	No	Yes	?	USA	*A. suspensa*	[148]
	Yes	No	Yes	Yes	USA (H)	*Z. cucurbitae B. latifrons*	[21,69]
**Figitidae**							
*Aganaspis daci*	No	No	Yes	?	COR	*C. capitata*	[163]
	No	No	Yes	?	COL	*C. capitata*	[139]
	No	No	Yes	?	MEX	*A. ludens A. obliqua*	[176]
	No	No	Yes	No	USA	*A. suspensa*	[159]
	No	No	Yes	?	USA (H)	*C. capitata*	[120]
**Ichneumonidae**							
*Bathyplectes exiguus*	No	No	Yes	?	CAN	*R. meiiginii*	[112]
**Pteromalidae**							
*Pachycrepoideus vindemmiae*	Yes	No	Yes	Yes	ARG	*A. fraterculus*, *C. capitata*	[16]
	Yes	No	Yes	Yes	BRA	*A. fraterculus*	[177]
	Yes	No	Yes	Yes	COR	*C. capitata*	[163]
	Yes	No	Yes	Yes	MEX	*A. serpentina*	[52]
	Yes	No	Yes	Yes	USA (H)	*Z. cucurbitae C. capitata*	[69,172]

Countries: ARG, Argentina; BEL, Belize; BOL, Bolivia; BRA, Brazil; CAN, Canada;, COL, Colombia; COR, Costa Rica; CUB, Cuba; DOR, Dominican Republica; ECU, Ecuador; ELS, El Salvador; FRG, French Guiana; GUA, Guatemala; MEX, Mexico; NIC, Nicaragua; PAN, Panama; PER, Peru; PUR, Puerto Rico; TRT, Trinidad and Tobago; VEN, Venezuela; USA (C), United States of America, California; USA (F), United States of America; USA (H), United States of America, Hawaii.

**Table 7 insects-11-00662-t007:** Guilds of introduced parasitoids of tephritid fruit flies in the Americas and Hawaii.

Introduced Parasitoid Species	Parasitism Modes	Feeding Types	Host Stage Attacked
**Braconidae**			
*Bracon celer*	S, K	En	L3
*Diachasmimorpha fullawayi*	S, K	En	L3
*D. krausii*	S, K	En	L2-L3
*D. longicaudata*	S, K	En	L2-L3
*D. tryoni*	S, K	En	L3
*Fopius ceratitivorus*	S, K	En	E-L1
*F. caudatus*	S, K	En	E
*F. arisanus*	S, K	En	E
*F. persulcatus*	S, K	En	L1
*F. silvestrii*	S, K	En	L2-L3
*F. vandenboschi*	S, K	En	L1
*Psyttalia concolor*	S, K	En	L2-L3
*P. cyclogaster*	S, K	En	L?
*P. fletcheri*	S, K	En	L3
*P. humilis*	S, K	En	L3
*P. incisi*	S, K	En	L3
*P. lounsburyi*	S, K	En	
*P.* *perproxima*	S, K	En	L2-L3
*P. ponerophaga*	S, K	En	L2-L3
*P. rhagoleticola*	S, K	En	L2-L3
*Utetes africanus*	S, K	En	L2-L3
**Chalcididae**			
*Dirhinus anthracia*	S, I	Ec	P
*D. giffardii*	S, I	Ec	P
**Diapriidae**			
*Coptera silvestrii*	S, I	En	P
**Eulophidae**			
*Aceratoneuromyia indica*	G, K	En	L3
*Tetrastichus giffardianus*	G, K	En	L3
**Figitidae**			
*Aganaspis daci*	S, K	En	L2-L3
**Ichneumonidae**			
*Bathyplectes exiguus*	U	U	L
**Pteromalidae**			
*Pachycrepoideus vindemmiae*	S-G, I	Ec	P

S, Solitary; G, Gregarious; I, Idiobiont; K, Koinobiont; Ec, Ecotoparasitoid; En, Endoparasitoid; L, Lavae; L1, Lava fisrt instar; L2, Larva second instar; L3, Larva third instar; P, Pupae; U, Unknown.

**Table 8 insects-11-00662-t008:** Predators of fruit flies in the Americas and Hawaii.

Predator	O	Ps	Prey	C	Reference
**Araneae**					
**Araneidae**					
*Argiope* sp.	**N**	A	*Z. curcubitae*	USA (H)	[178,179]
**Lycosidae**					
*Trochosa terricola*	N	A	*R. pomonella*	CAN	[180]
**Philodromidae**					
*Philodrornus praelustris*	N	A	*R. pomonella*	CAN	[181]
*Philodromus vulgaris*	N	A	*R. pomonella*	CAN	[181]
**Salticidae**					
*Eris militaris*	N	A	*R. pomonella*	CAN	[181]
*Megafreya sutrix*	N	A	*A. fraterculus*	BRA	[182,183]
*Paraphidippus aurantius*	N	A	*A. ludens*	MEX	[184]
*Pelegrina proterva*	N	A	*R. pomonella*	CAN	[181]
*Phidippus audax*	**N**	A	*A. ludens*	MEX	[185]
*Phidippus bidentatus*	**N**	A	*A. ludens*	MEX	[184]
*Salticus scenicus*	N	A	*Rhagoletis* sp.	CAN	[69]
**Acari: Macrochelidae**					
*Macrocheles* sp.	N	A	*A. suspensa*	CUB	[186]
*Macrocheles roquensis*	N	A	*C. capitata*	BRA	[187]
**Orthoptera**					
**Gryllidae**					
*Gryllus pennsylvanicus*	N	L.P	*R. pomonella*	CAN	[180,188]
*Nemobius fasciatus*	N	L.P	*R. pomonella*	CAN	[188]
**Trigonidiidae**					
*Allonemobius fasciatus*	N	L.P	*R. pomonella*	CAN	[180]
**Dermaptera**					
**Anisolabididae**					
*Euborellia annulipes*	N	L.P	*A. suspensa*	USA	[189]
**Chelisochidae**					
*Chelisoches morio*	**E**	L	*Z. curcubitae*	USA (H)	[178,179]
**Coleoptera**					
**Carabidae**					
*Amara aenea*	N	L, P	*R. pomonella*	CAN	[180]
*Calosoma calidum*	N	L.P	*R. pomonella*	CAN	[180]
*Carabus nemoralis*	N	L	*R. mendax*	CAN	[190]
*Harpalus aeneus*	N	L, P	*R. pomonella*	CAN	[180]
*Harpalus pennsylvanicus*	N	L.P	*R. pomonella*	CAN	[180]
*Poecilus lucublandus*	N	L	*R. mendax*	CAN	[190]
*Pterostichus melanarius*	E	L, P	*R. pomonella*	CAN	[180]
		L, P	*R. mendax*	CAN	[191]
*Pterostichus mutus*	N	L, P	*R. mendax*	CAN	[190]
**Elateridae**					
*Conoderus* sp.	N	P	*A. suspensa*	USA	[189]
**Staphylinidae**					
*Belonochus rufipennis*	N	L	*A. fraterculus*	BRA	[192]
		L	*A. ludens*	MEX	[193]
		L	*C. capitata*	BRA	[192]
*Dinothenarus badipes*	N	L, P	*R. pomonella*	CAN	[180]
*Homaetarsus* sp.	N	L	*A. ludens*	MEX	[194]
*Philonthus* sp.	N	L, P	*R. pomonella*	CAN	[180]
*Philonthus turbidus*	E	L	*Z. curcubitae*	USA (H)	[178,179]
**Hemiptera: Reduviidae**					
*Zelus renardii*	E	A	*Z. curcubitae*	USA (H)	[178,179]
**Hymenoptera: Formicidae**					
*Crematogaster* sp.	N	L	*A. ludens*	USA	[194]
*Dorymyrmex* sp. 1	N	L	*Anastrepha* sp.	BRA	[23]
*Ectatomma brunneum*		L	*Anastrepha* sp.	BRA	[23]
*Formica fusca*	N	L, P	*R. pomonella*	CAN	[180]
*Leptothorax* sp.	N	L, P	*A. suspensa*	USA	[189]
*Monomorium* sp.	N	L, P	*A. suspensa*	USA	[189]
*Myrmica* sp.	N	L, P	*R. pomonella*	CAN	[180]
*Pachycondyla striata*	N	L	*A. fraterculus*	BRA	[25]
*Paratrechina parvula*	N	L, P	*A. suspensa*	USA	[189]
*Pheidole gertrude*	N	L	*Anastrepha* sp.	BR	[23]
	N	L	*A. ludens*	MEX	[193]
*Pheidole magacephala*	E	E, L, P	*Z. curcubitae*	USA (H)	[178,179]
*Pheidole oxyops*	N	L	*Anastrepha* sp.	BRA	[23]
*Pogonomyrmex naegelli*	N	L	*A. fraterculus*	BRA	[25]
*Odontomachus brunneus*	E	L, P	*A. suspensa*	USA	[189]
*Solenopsis geminata*	N	L	*A. ludens*	MEX	[22,193]
	E	L	*A. ludens*	USA	[194]
	E	E, L, P	*Z. curcubitae*	USA (H)	[178,179]
	N	L	*C. capitata*	GUA	[195]
	E	P	*C. capitata*	USA	[69]
*Solenopsis invicta*	E	L	*A. suspensa*	USA	[189]
*Solenopsis saevissima*	N	L	*A. fraterculus*	BRA	[25]
**Reptilia: Polychrotidae**					
*Anolis serranoi*	N	L	*A. ludens*	MEX	[196]
**Rodentia: Cricetidae**					
*Peromyscus leucopus*	N	P	*A. ludens*	MEX	[197]
*Peromyscus boylii*	N	P	*A. ludens*	MEX	[197]

O, Origin; N, Native; E, Exotic; C, Country; PS, Pray stage attached PS; A, Adult; E, Egg; L, Larvae; P, Pupae; BRA, Brazil; CAN, Canada; CUB, Cuba; GUA, Guatemala; MEX, Mexico; USA, United States of America; USA (H), United States of America, Hawaii.

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
