# Peer review of "Biological Control of Tephritid Fruit Flies in the Americas and Hawaii: A Review of the Use of Parasitoids and Predators"

_insects, 2020, doi:10.3390/insects11100662_

Round 1

Reviewer 1 Report

The manuscript is a very comprehensive review of the biological control programs undertaken over many years in the Americas against very serious agricultural pests. As such is it a very valuable historical document  on the one hand and will be of great assistance to workers in this field world wide, on the other. My only complaints are inconsistensies with spelling of certain words (Mexico for example, authors of genera added or not, commas in the tables) and errors of language, most of which I have spotted and indicated, but I may have overseen others as there are so many. I do not mind my name being known to the authors.

Author Response

  • We want to thank the reviewer for the suggestions and corrections made in our manuscript.
  • We have made all the corrections suggested by the reviewer.
  •  

Reviewer 2 Report

The paper deals with a long review of biological control of Fruit Flies in Hawaii island and in the Americas. The paper is well written and really interesting and give useful information for scientists as well for technicians involved in biological control programs of FF. I know that there is huge amount of literature in this field and the attempts to summarize all the actions has produced too much long tables which are not so much readable, but I understood the need to have done them this way. There are some mistakes that I reported in a separate file.

My opinion is that the present paper is ready for publication after a few minor revisions.

line 95: externallythey

line 97: substitute on with of

line 108: Bactrocera instead of Batrotera

line 113: the same

line 115: genera instead of genus

line 128: Ceratitis instead of Cerattis

lines 157-161: sentences are not so clear, please clarify

line 201: Staphilinidae instead of Stathyllinidae

lines 448 – 459: these last sentences close the review in an inelegant and even a little misleading way; I recommend inserting them before or even taking them out of context

Author Response

  • We want to thank the reviewer for the suggestions and corrections made in our manuscript.
  • We have made all the corrections suggested by the reviewer.

Reviewer 3 Report

The manuscript insects-934273 (Biological control of tephritid fruit flies in the Americas and

Hawaii: A review of the use of parasitoids and predators) is a nice well written manuscript provides highlights and important data on parasitoids and predators and reviewing recent biocontrol programs in the Americas and Hawaii. The manuscript will facilitate the knowledge for researchers working on biocontrol of frugivorous tephritid fruit flies. I added my notes and concerns on the attached pdf file for authors consideration during revision. I agree on publication in Journal of Insects with minor revision and my main concerns are:

  • The rationale for including Hawaii, a region of Oceania realm, to the Americas for this review is not discussed. Is it because Hawaii is a State of USA?
  • Throughout the manuscript I argued that the terms egg-pupal (as in F. arisanus), larval-pupal (as in Diachasmimorpha spp. and Psyttalia spp), and larval-prepupal should be reverted to the original older terms (egg-larval and larval parasitoids). Simply because the parasitoid will arrest the host development and allows the host to pupariate but the host inside the puparium will never turn to a pupal form (with legs, wing pads, head, thorax, and abdomen). Only this happened with Aganaspis spp., as true larval-pupal parasitoids. Also, the prepupa is a physiological state of the last larval instar.
  • Check synonyms, Spalangia nigra and S. hirta. First, they are not native to Hawaii, and S. hirta is a synonym for S. nigra according to Universal Chalcidoidea.
  • There is no native frugivorous fruit flies nor native fruit fly parasitoids in Hawaii. Please correct that with footnotes in the table titles which read (Native parasitoid species and their fruit fly’s hosts in the Americas and Hawaii).
  • Listing B. cucurbitae as a host for many opiines (D. tryoni, D. longicauata, F.arisanus, F. vandenboschi) and the eulophid (T. giffardianus) is misleading to readers who want to select natural enemies for this fruit fly. Only Psyttalia fletcheri on the list is a natural parasitoid for this resistant pest. Also, use the new genus name Zeugodacus.
  • In the abstract the authors listed Hawaii for predators, but the table has no representative for Hawaii' introduced predators. For example, Thyrecephalus albertisi (Staphylinidae) was introduced from Philippines in 1947 for biocontrol of medfly. This rove beetle is established on all islands. Other rove beetles introduced for Carpophilus spp. (Nitidulidae) also attack fruit flies in the infested fallen fruits (Philothapus analis).
  • If Anastrepha grandis has no parasitoids, what is controlling the populations of this fly in the Americas? Too bad, I was hopeful that you report some opiines for use against Z. cucurbitae in Hawaii. Can you speculate why?

Author Response

  • We want to thank the reviewer for the suggestions and corrections made in our manuscript.
  • We made most of the corrections suggested by the reviewer.
  • Yes, we have included Hawaii as a state of the United States. We consider it important to add information about Hawaii because it is important for the economy of one of the largest countries in America and because it has important examples of biological control programs for fruit flies.
  • All old terms have been replaced by new ones throughout the whole text.
  • We excluded the species S. hirta and transferred the information to S. nigra
  • The word “Hawaii” was removed from all the information about native parasitoids
  • We used the genus Zeugodacus when it was necessary (Z.  cucurbitae)
  • .Although Psyttalia fletcheri is the most effective parasitoid in the control of Z.  cucurbitae, there are other parasitoid species mentioned in the literature that parasitize this species. As our Article is a review we cannot omit these works,
  • The predators of Hawaii fruit flies found in the literature consulted were included;
  • We have included a paragraph to explain this fact in lines 141 to 146.

Reviewer 4 Report

A good, comprehensive review.  Use the term "conservation" instead of "conservative" for conservation biological control.

Line 93: solitary, gregarious - remove "more than one", but in fact this paragraph is not needed.  There is no need to define these terms for the reader.

Line 107: please clarify that there are native parasitoids that are attacking introduced species of fruit flies.

Line 113 and 115:  use "genera" (=plural form) not "genus" (=singular form).

Line 121: use recorded not registered

Line 122:  use host not hosts

Line 129: rewrite to "no associated parasitoids"

Line 134: rewrite to "continental USA"

Line 160:  Here there is the use of the umlaut over the 'o' (ö) in Forster, but in other places in the manuscript it is spelled Foerster.  Make it consistent

Line 166: use most instead of more

Line 189: use introduced, koinobionts - no 'e'

Line 203: - remove comma

Line 204: spider's

Line 204: - do you mean larvae and/or pupae?

Line 302: eradicated

Line 301: missing a 'b' in biological

Author Response

(The authors gave the same response as above.)
